# The Interaction of Waterborne Epoxy/Dicyandiamide Varnishes with Metal Oxides

**DOI:** 10.3390/polym14112226

**Published:** 2022-05-30

**Authors:** Gary Säckl, Jiri Duchoslav, Robert Pugstaller, Cornelia Marchfelder, Klaus Haselgrübler, Maëlenn Aufray, David Stifter, Gernot M. Wallner

**Affiliations:** 1Christian Doppler Laboratory for Superimposed Mechanical-Environmental Ageing of Polymeric Hybrid Laminates (CDL-AgePol), Johannes Kepler University Linz, Altenberger Straße 69, 4040 Linz, Austria; robert.pugstaller@jku.at (R.P.); cornelia.marchfelder@jku.at (C.M.); gernot.wallner@jku.at (G.M.W.); 2Center for Surface and Nanoanalytics (ZONA), Johannes Kepler University Linz, Altenberger Straße 69, 4040 Linz, Austria; jiri.duchoslav@jku.at (J.D.); klaus.haselgruebler@jku.at (K.H.); david.stifter@jku.at (D.S.); 3Institute of Polymeric Materials and Testing (IPMT), Johannes Kepler University Linz, Altenberger Straße 69, 4040 Linz, Austria; 4Centre for Electrochemical Surface Technology (CEST), Viktor Kaplan Straße 2, 2700 Wiener Neustadt, Austria; 5CIRIMAT, Universite de Toulouse, CNRS, INP-ENSIACE, T 4 allée Emile Monso-BP44362, CEDEX 4, 31030 Toulouse, France; maelenn.aufray@ensiacet.fr

**Keywords:** XPS, epoxy, DICY, varnish, coating, interface, interphase, ultra-low-angle microtomy

## Abstract

For delayed crosslinking of waterborne epoxy varnishes, dicyandiamide (DICY) is often used as a latent curing agent. While, for amine-based curing agents such as diaminoethane (DAE), chemical interactions with metal oxides are well described, so far, no studies have been performed for DICY and waterborne epoxy varnishes. Hence, in this work X-ray photoelectron spectroscopy (XPS) was used to investigate reactions of DICY and varnishes with technical surfaces of Al, Zn, and Sn. To directly study the reaction of DICY with metal oxides, immersion tests in a boiling solution of DICY in pure water were performed. A clear indication of the formation of metal–organic complexes was deduced from the change in the N1s peak of DICY. To understand the interfacial interaction and consequently the interphase formation during coating of waterborne epoxy varnishes, advanced cryo ultra-low-angle microtomy (cryo-ULAM) was implemented. Interestingly, a comparable reaction mechanism and the formation of metal complexes were confirmed for varnishes. The coatings exhibited a pronounced enrichment of the DICY hardener at the metal oxide–polymer interface.

## 1. Introduction

Epoxy-based varnishes are used as protective coatings, adhesives, or encapsulants [1,2,3,4,5,6]. For crosslinking of epoxy resins, which are often based on the Diglycidyl ether of bisphenol-A (DGEBA), diamines are well established. For high-temperature adhesives and coating systems, dicyandiamide (DICY) is of high relevance due to its latent curing characteristics [7,8]. If epoxy–diamine liquid mixtures are applied to metal substrates, metal–organic complexes or chelates are formed due to reactions of amines with metal oxides [9,10]. As a consequence, an interphase with properties differing from the bulk material of the polymeric adhesive is generated. Thicknesses of the interphase of several hundred micrometers have been reported [10]. Such coating–substrate reactions have been studied in detail for different amine curing agents, such as diaminoethane (DAE). However, no systematic investigations have been performed for DICY and various metals. X-ray photoelectron spectroscopy (XPS) has already been successfully applied to confirm such interactions of, e.g., DETA with metal surfaces [11,12,13]. For DICY, a chemical interaction with Al oxide surfaces was ascertained [7].

The main objective of this work was to study the reaction of DICY with various metal oxides by both simple immersion testing of metals in DICY–water solutions and application of a waterborne epoxy varnish on metal substrates. In contrast to other studies [7,9,10,11,12,13], the investigations were focused on technical surfaces of Al, Zn, and Sn sheets. Furthermore, the waterborne epoxy varnish was applied under ambient conditions and subsequently cured at 200 °C. The metal sheets, the individual components of the epoxy model varnish [14], and the cured coatings were characterized by XPS, which allows for a unique vertical sensitivity and a shallow information depth of under 10 nm. Depth profiling by, e.g., Ar ion sputtering is not feasible for obtaining chemical information on organic materials due to the destructive nature of the sputtering process [15]. As an alternative, cryo ultra-low-angle microtomy (cryo-ULAM) has been proven to be capable of removing parts of polymeric coatings without destroying the underlying chemistry [15,16]. Hence, specific focus was placed on cryo-ULAM and the analysis of the metal–polymer interface.

## 2. Materials and Methods

### 2.1. Materials and Sample Preparation

Sheets of pure metal (Al, Zn, and Sn) with thicknesses ranging from 0.5 to 0.8 mm were purchased from Polymet (Dietzenbach, Germany). For XPS analysis, the sheets were cleaned with tetrahydrofuran and subsequently with isopropanol in an ultra-sonic bath for 30 min. Each cleaning step was performed two consecutive times.

For immersion testing, the sheets with an original size of 170 × 200 mm^2^ were cut into pieces with a size of 5 × 5 mm^2^. Pure DICY powder (Sigma-Aldrich, Vienna, Austria) was diluted in water up to a concentration of 3 wt% under stirring. Water was chosen as a solvent for DICY since it is also a component of waterborne epoxy coatings. Furthermore, to mimic the elevated temperature during curing, the solution was brought to boiling temperature and the metals were then immersed for 30 min, as schematically shown in Figure 1a. After immersion, the samples were rinsed with pure water and transferred to the XPS chamber after drying in air. As a reference, the same procedure was also carried out with pure boiling water.

The epoxy resin used for the formulation of the waterborne epoxy model varnishes was purchased from Olin Corporation (Clayton, MO, USA). The epoxy equivalent weight (EEW) ranged from 860 to 930 g/mol. For stabilization, an emulsifier based on a poloxamer was added, with a relative concentration of approximately 10 wt% with respect to the primary epoxy content. The epoxy and poloxamer dry blend were dispersed in 45 wt% water. As crosslinking agent, DICY was used with a chosen over-stoichiometric ratio of 1.1 to 1.0 (DICY to epoxy). More detailed information on the formulation and processing of the waterborne epoxy model varnish is given in [14]. The varnish was applied with a drawbar onto metal sheets and then pre-cured, resulting in an average coating thickness of approximately 5 µm. In a final step, the coatings were fully cured at 200 °C for 2 h under a N_2_ atmosphere.

A Leica Ultracut UCT ultramicrotome with a standard glass knife was employed. The coated metals were cut into pieces with a size of 4 × 12 mm^2^ and glued onto a customized aluminum sample holder. The sample holder was slightly tilted with an angle ≤3°, providing a wedge-shaped section, which revealed the underlying coating material and the metal surface as depicted in Figure 1b. During the cutting process, the sample and knife were cooled in a chamber purged with liquid nitrogen to a temperature of 0 °C (cryo-ULAM). As successfully proven for other polymeric coatings [16], cooling allows for minimization of smearing effects. For optical and scanning electron microscopy (SEM), the obtained tapers were sputter coated with a 20 nm thick gold layer using a CCU-010 high vacuum sputter coating system (Safematic, CH, Zizers, Switzerland).

### 2.2. Analytical Characterization

A LEXT OLS4000 laser confocal microscope was used to examine the topography of the metal substrates (Olympus, Hamburg, Germany). The tapers produced by cryo-ULAM were also characterized using a ContourGT-K optical profiler (Bruker, Rheinstetten, Germany). Furthermore, SEM investigations were performed on a JSM6400 system (Jeol, Tokyo, Japan) with a working distance of 39 mm and an accelerating voltage of 15 kV. The XPS apparatus used in this study was a Theta Probe system (Thermofisher, Loughborough, UK). The acquisition and evaluation of the data were performed with the Avantage software package provided by the system’s manufacturer. A monochromatic Al Kα X-ray source (1486.6 eV) was employed. For investigation of the materials in the reference state and the immersed metal samples, a spot size of 400 µm was selected. For linescans on ULAM cuts, a smaller spot size of 150 µm and a step size of 50 µm were chosen. To compensate for charging effects during acquisition, an FG02 dual flood gun (Thermofisher, UK) was used to neutralize the region of interest with low-energy electrons and Ar ions. The spectra were charge-corrected by referencing the C-C/C-H peak to 285.0 eV. Furthermore, for data evaluation the background of the photoelectron peaks was subtracted using the so-called smart background function of the Avantage evaluation software. For quantification of the elemental composition, the area under the peaks was normalized using Scofield sensitivity factors. For fitting of high-resolution (HR) spectra, a product of a Lorentzian and Gaussian-shaped peak was used.

## 3. Results and Discussion

### 3.1. Chemical Structure of Components for the Waterborne Epoxy Varnish

The main components of the model varnish were analyzed by XPS. The epoxy resin (DGEBA), which is based on O and C, showed two different peaks in the C1s HR spectrum (see Figure 2a). While the peak at 285.0 eV was assigned to C-C (and C-H) bonds, the peak that shifted by 1.8 eV to a higher binding energy was related to C-O bonds. The number n of repeating units was nominally 4. Therefore, C-O-C and C-OH groups dominated over the epoxy end groups, which reportedly exhibit a shift of 2.0 eV [17]. Consequently, differentiation between epoxy and the other C-O bonds was difficult. The binding energy of the O1s peak at 533.2 eV was equally assigned to C-O bonds [17]. Furthermore, in Figure 2b HR spectra of the emulsifier used for the model varnish are depicted. Two C1s peaks and one O1s peak were ascertained. However, the peak positions varied slightly compared with the epoxy resin (see Table 1). The binding energy shift of 1.4 eV of the C1s C-O bonds was comparable to that of other C-O bond-containing polymeric materials [17]. The O1s peak revealed a binding energy of 532.6 eV, which was lower compared with the epoxy resin. Moreover, the intensity of the three peaks was different (see Table 1). These slight differences are presumably related to aromatic–aliphatic C-O-C bonds in epoxies, whereas the emulsifier exhibits aliphatic C-O-C bonds only.

In Figure 2c, the C1s spectrum of DICY is depicted. Three different peaks are discernible. The weak C-C peak is most likely related to slight contamination. The strong peak at 287.0 eV can be attributed to C bound to the nitrile group [18]. The C bound to the three N atoms had a higher energy shift of 288.9 eV. The intensity of both peaks was comparable. In the asymmetric N1s spectrum, two peaks could be fitted. The nitrogen atoms of the amine and nitrile had approximately the same binding energy and were thus combined into one peak at 399.9 eV [19,20,21]. The N in the imine was assigned to a lower binding energy of 398.6 eV [18,19,21]. The ratio of the two peaks was in good agreement with the chemical structure of DICY. A I_398.6eV_: I_399.9eV_ ratio of 0.35 was obtained, which is close to the theoretical value of 0.33, resulting from the DICY molecule having one imine group, two amine groups, and one nitrile group.

### 3.2. Metal Oxides and Their Interaction with DICY in an Aqueous Enviroment

Since the model varnish was water-based, the chemical interaction of DICY and the metal surfaces was investigated and mimicked by immersion in a boiling water environment. Pure water (for reference) and the solution of DICY were brought to boiling and the Al, Sn, and Zn specimens were immersed for 30 min. The specimens were investigated before and after these treatments. As mentioned above, the metal sheets exhibited technical surfaces. The arithmetic surface roughness Ra and the maximum valley height Rz for the different metal sheets varied slightly. As measured by confocal microscopy, the following roughness values were deduced considering a cut-off wavelength of 250 µm:
Al: Ra = 0.47 µm; Rz = 4.11 µmZn: Ra = 0.25 µm; Rz = 3.12 µmSn: Ra = 0.26 µm; Rz = 6.75 µm

The roughness values of Al and Sn after the immersion treatments were comparable to the reference. The roughness of the Zn surfaces increased significantly after immersion in pure water (Ra = 0.43 µm and Rz = 19.98 µm) and in the DICY solution (Ra = 0.35 µm and Rz = 12.00 µm). Moreover, the Zn substrate revealed an inhomogeneous brownish color after immersion in the DICY solution.

#### 3.2.1. Effect of Immersion on XPS Spectra of Metal Substrates

In Figure 3a–c, the XPS HR spectra of the corresponding metals are depicted before and after immersion in pure water and in the DICY–water solution. Since the chemical shift in the metallic and oxidic Zn was too small to be separated for the Zn2p peak, the LMM Auger signal was considered. For all metals, the treatment in pure water or in the solution of DICY in water resulted in an increase in the thickness of the oxide layer. The corresponding metal peaks were not discernible after immersion treatments. The respective oxide layer thicknesses exceeded the information depth of XPS. Moreover, the peak width and binding energy of the oxidic components of Al and Sn decreased slightly after immersion in the boiling liquids. Hence, a change in the chemical structure was indicated, associated with a hydroxylation of the oxide layers.

For Zn, a difference between the treatment with pure water and with the DICY solution was observed. A detailed analysis of the chemical changes can be accomplished by determining the modified Auger parameter [22]. The obtained modified Auger parameter before and after immersion was compared with references for ZnO and Zn(OH)_2_. In the reference state and after immersion of Zn in boiling water, modified Auger parameters of 2010.0 eV and 2009.9 eV were deduced. These values are comparable to findings reported in the literature for ZnO (2009.9 eV) [22]. After immersing the specimen in the boiling DICY solution, the modified Auger parameter changed to 2009.1 eV, which is representative for Zn(OH)_2_ (2008.9 eV) [22]. Since, for the Zn specimen immersed in pure water, no change was observed for the oxidic component in the LMM spectrum, just the spectrum for the sample boiled in the DICY–water solution is illustrated and compared to pure DICY in Figure 4. Clear indications of interaction of the hardener DICY with the Zn oxide surface are discernible. After the dissolution of Zn at the surface in contact with the DICY solution, the following reactions are proposed to occur:H_2_O ⇌ H^+^ + OH^−^(1)
-NH_2_ + H^+^ ⇌ -NH_3_^+^(2)
Zn^2+^ + 4OH^−^ ⇌ Zn(OH)_2_ + 2OH^−^ ⇌ [Zn(OH)_4_]^2^^−^(3)

For Al and Sn, the reaction might look similar; however, the modified Auger parameter was not accessible for Al with the used X-ray source. Furthermore, references for Sn-related compounds are not available. In agreement, the formation of metal–organic complexes by interactions of diamines and metal ions was reported in [9,10,12,13,23,24].

#### 3.2.2. Effect of Immersion on DICY Residues on Metal Substrates

For Al, Zn, and Sn, chemical changes in DICY residues were observed due to boiling in the DICY–water solution. However, for Al the intensity of the N1s peak was very low (<0.6 at%). In contrast, the N content was 5.7 at% and 8.8 at% for the Zn substrate and the Sn substrate, respectively. N1s spectra of Zn and Sn metal sheets immersed in the DICY solution are depicted in Figure 4b,c. Comparing the spectra with the DICY reference (see Figure 4a), more symmetry was discernible for the DICY residues on the metal substrates. Moreover, the width of the peak had increased, most noticeably for DICY, which reacted with the Sn oxide layer. Hence, an additional peak was fitted at a higher binding energy. To ensure that the fit was consistent, the position and the full width at half maximum (FWHM) of the two peaks from the reference spectra were used. The additional peak at 400.6 eV can be either assigned to a protonated amine or a hydrogen bond with a metal hydroxide [11,18,21]. A similar interaction was proposed for DAE reacting with Zn oxide [11]. The hydroxide acts as a Brønsted acid. Additionally, the ratio between the two peaks of amine/nitrile and imine functionalities changed drastically. A new peak appeared with a binding energy comparable to that of the imine. This is presumably related to Lewis bonding of the metal ions to DICY, as already suggested for Al and Zn in [11,12,13]. The metal cation can either act as a Lewis base or acid. The binding energy of such a Lewis bond might lie between the imine and amine contributions. A value of 399.4 eV was reported for Zn reacted with amine [11]. Furthermore, a similar interaction could also be possible for the nitrile functionality of DICY, which may also form a Lewis bond with metal cations associated with similar binding energies [7]. The deconvolution of these peaks is difficult. Hence, the peak at 398.6 eV was assigned to the imine and the Lewis bond of DICY with metal cations, marked as the α state in Figure 4. The possible resulting products and interactions are depicted in Figure 4d.

The chemical reactions were different for the investigated metals. For Al, the reaction might be weak compared with Zn and Sn, since after rinsing there was only a small amount of N left on the Al surface. Furthermore, the N1s peaks for DICY reacted with Sn and Zn were different to each other. The ratio of the higher binding energy peak of the amine/nitrile at 399.9 eV and the lower binding energy peak at 398.6 eV was 1.08 and 0.54 for Zn and Sn, respectively. That was significantly higher than the ratio of 0.35 obtained for the DICY reference. Moreover, the fitted peak feature at 400.6 eV was higher for DICY reacted with Sn in comparison with Zn. As reported in [9,10,12,13,23,24], the formation of organic–metallic complexes varies depending on the metallic substrate’s nature. The results of this study support these findings. The interactions between DICY and the investigated metals (Al, Zn, and Sn) were quite different.

### 3.3. Interphase Formation during Coating of Metal Substrates with Waterborne Epoxy Varnishes

A chemical interaction of DICY with the metal oxide surfaces, directly applied in the immersion tests, was confirmed in Section 3.2. In the following, SEM and ULAM-XPS results are described for micrometer-thin, cured epoxy coatings on metal substrates.

#### 3.3.1. Structural Features of ULAM Cuts

SEM images of a coated sample cut by cryo-ULAM are depicted in Figure 5. The coating and polymer–metal interface can be distinguished by the Z-contrast, since the metal will have a higher backscattered electron yield and appears, therefore, brighter in the image. The exposed bulk of the coating, resulting from the cutting process, can be recognized by a brighter contrast compared with the original surface. The bulk of the thin coating was termed and is marked as an interphase. The length of the cut was up to several hundred micrometers. In Figure 5, a magnification of the transition from the interphase to the polymer–metal interface is depicted. The detailed image clearly reveals strip-like structures, which are presumably related to epoxy residues smeared by the glass knife into the valleys and grooves of the microrough technical surface of the metal sheets. Furthermore, the glass knife might also deform the surface of the metal substrate.

In Figure 6, the same ULAM cut is depicted as a 3D image acquired by white light interferometry. The original coating’s surface was wavy, and the thickness varied slightly. As evaluated from the linescan, the angle of the cut was 3.0°. During microtomy, the underlying metal substrate was cut or deformed, since the depth significantly exceeded the thickness of the coating, which was around 5 µm. Considering the obtained angle, the taper shown in Figure 5 is realistic and sufficient for obtaining several XPS linescan positions at a spot size of 150 µm.

#### 3.3.2. Elemental Composition of Epoxy Coatings on Metal Substrates

Immediately after ULAM-cutting, the samples were transferred to the XPS vacuum chamber. In Figure 7, XPS linescans over tapers of coated Al, Zn, and Sn are depicted. At an X-ray spot size of 150 µm, a step size of 50 µm was chosen for the scans. A clear difference in the elemental composition was obtained for the surface of the coating, the uncovered interphase, and the metal–polymer interface. For Al and Zn, the N content on the coating’s surface was below the detection limit. In contrast, N was detectable on the surface of the coating on the Sn. However, the content was slightly lower compared with the uncovered interphase. The low content of DICY on the coating’s surface is presumably related to segregation phenomena driven by differences in the size of DGEBA and DICY molecules [25]. Considering only epoxy and DICY, the nominal concentration of N should be 0.8 at%. Comparable values were measured in the interphase region of the coatings on the Al, Zn, and Sn.

The HR spectra of the C1s peak (not shown) along with the linescans were comparable to the spectra of the epoxy resin presented in Section 3.1. Furthermore, the overall concentration of O and C was in agreement with the epoxy oligomer. However, differences were discernible at the interface of the metal and the polymer. As discussed in the previous subsection, the glass knife also cut and deformed the metal. However, the original surface roughness as well as the grooves induced by the glass knife led to thin epoxy residues on the uncovered metal substrate. Interestingly, the N content increased at this uncovered interface just for Zn and Sn (see Figure 7b,c). For Al, the N content was almost constant, even though the relative amount of C decreased significantly. Consequently, for the investigated metals a relative enrichment of DICY at the interface was confirmed. The enrichment of DICY at the interface was pronounced for Zn and even more so for Sn. Segregation of an amine-based hardener near the surface was also reported for Cu substrates [26]. While the segregation of the DICY hardener varied for the different metals, the concentration of N and, consequently, the concentration of DICY in the uncovered interphase of the coatings on the different metals were comparable.

#### 3.3.3. Chemical Changes near the Interface

While the C1s HR peaks of the coating’s surface, interphase, and interface regions were comparable to the reference spectrum of the epoxy oligomer, a clear change in chemistry was discernible in the N1s spectra. In Figure 8a–c, the HR spectra of the N1s peak determined for the uncovered interphase region and the interface with the metal are depicted. The concentration of N was quite low. Nevertheless, a dominant amine/nitrile peak at 399.9 eV was detected in the uncovered interphase. Hence, not all of the functional groups were reacting during double-stage curing. Similar to the reference spectra of DICY, a small peak contribution at a binding energy of 398.6 was ascertained. At the interface of the metal and epoxy coating, the N1s peak became more symmetric. A comparable effect was observed after immersion of the metals in the DICY–water solution. The peak at 400.6 eV associated with hydrogen bonds or protonated amine was more pronounced for Zn and Sn. Furthermore, the intensity at the lower binding energy close to the imine peak at 398.6 eV was higher at the interface, especially for Zn and Sn. However, this peak revealed a slightly higher binding energy than imine (398.6 eV). As reported in [11], it is presumably related to Lewis bonding of metal cations with the amine or nitrile. The N1s peak features, which have also been described for diamines [11,12,13,24], confirmed interactions of metal ions with diamines during application and double-stage curing of waterborne varnishes on Al, Zn, and Sn. Chemical reactions and the potential formation of metal–organic complexes are favored near the interface. Diffusion of such complexes into the bulk of the coating termed the interphase could not be ascertained unambiguously. The concentration of the imine or Lewis bonding of metal cations with amine or nitrile moieties was too low in the interphase. However, a peak presumably related to hydrogen bonds or protonated amine (close to 400.6 eV) was also detected in the interphase, especially for the epoxy–Sn sample.

## 4. Summary and Conclusions

The main objective of this study was to elucidate potential interactions of waterborne epoxy varnishes based on the latent curing agent DICY and Al, Zn, and Sn metal substrates. Therefore, analytical investigations in different material states were carried out. As a reference, the main components of the varnish formulation (i.e., epoxy resin, emulsifier, and DICY) and the surface of the metal substrate were characterized by XPS. Furthermore, the metal substrates were exposed to boiling water and a solution of DICY in water. Structural changes in the surface were examined. Finally, the epoxy varnish was applied to the substrates, pre-coated, and fully cured. To uncover the bulk of the coating, a potential interphase, and the epoxy–metal interface, the coated metals were cut by cryo-ULAM and characterized and scanned by XPS.

After immersion of the metal substrates in the DICY–water solution, clear indications of the formation of metal–organic complexes were discernible. The N1s peak of DICY residues on the metal substrate revealed further subpeaks attributable to protonated amine or hydrogen bonds and Lewis bonds of the metal ions with the amine and nitrile functionality of DICY, which were not detectable for DICY in the reference state. These structural changes were much more pronounced for Zn and Sn as compared with Al. Furthermore, immersion in boiling water or the aqueous DICY solution lead to an increase in the oxide layer’s thickness and a clear change of the Zn oxide to Zn(OH)_2_. XPS linescans of tapers prepared by cryo-ULAM and taken from coated Al, Zn, and Sn revealed an enrichment of DICY close to the interface. The enrichment was more pronounced for Zn and Sn. Furthermore, a quite similar reaction was confirmed for the coatings as well as the metals immersed in aqueous DICY solutions. Indications of the formation of metal–organic complexes at the interface of the epoxy coating and the metal surface were deduced. These findings on the investigated waterborne epoxy varnish coated onto metal substrates are in agreement with the epoxy amine–metal interphase formation concept established for different amine hardeners and water-free adhesives [9,10,12,13,23,24].

## Figures and Tables

**Figure 1 polymers-14-02226-f001:**
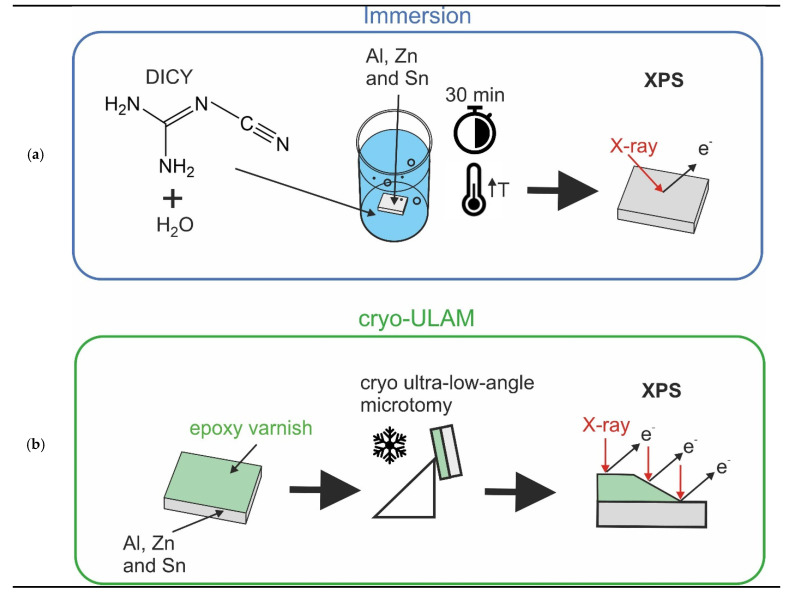
(**a**) Schematic representation of the immersion tests and (**b**) cryo-ULAM preparation followed by an XPS linescan.

**Figure 2 polymers-14-02226-f002:**
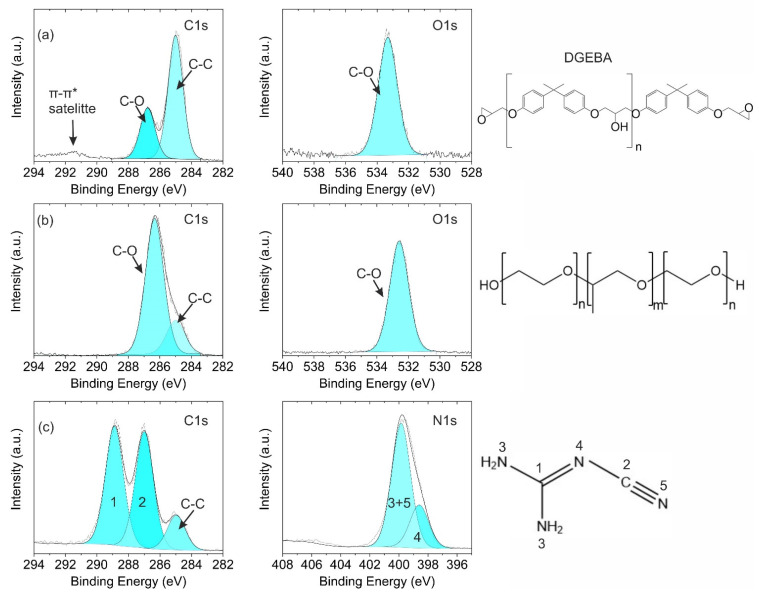
(**a**) C1s (left) and O1s (right) HR spectra of the epoxy resin, (**b**) C1s (left) and O1s (right) HR spectra of the emulsifier, and (**c**) C1s (left) and N1s (right) HR spectra of DICY.

**Figure 3 polymers-14-02226-f003:**
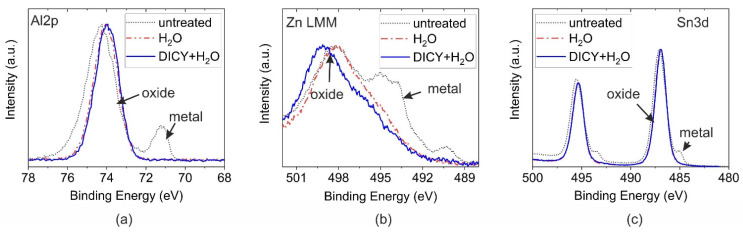
HR spectra of the metal photoelectric and Auger peaks before and after immersion in boiling water and the DICY–water solution for (**a**) Al, (**b**) Zn, and (**c**) Sn.

**Figure 4 polymers-14-02226-f004:**
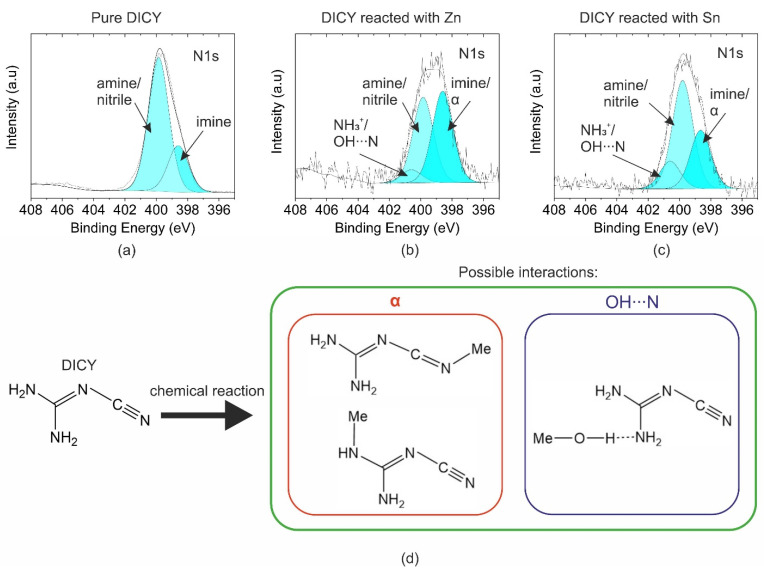
XPS N1s HR peaks of pure DICY (**a**), DICY residues on Zn (**b**), and DICY residues on Sn (**c**). The suggested chemical products and interactions are depicted in (**d**).

**Figure 5 polymers-14-02226-f005:**
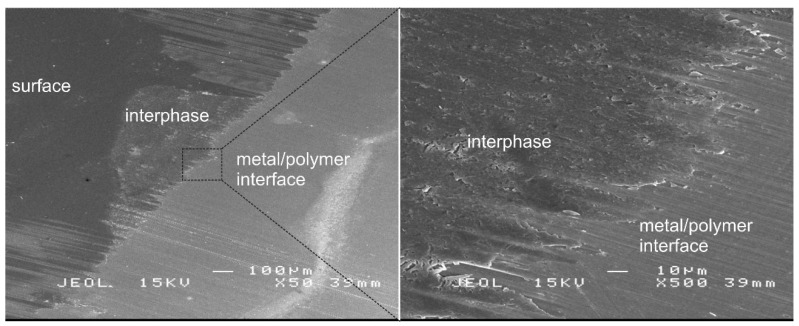
SEM images of a cryo-ULAM cut at magnifications of 50× (**left**) and 500× (**right**).

**Figure 6 polymers-14-02226-f006:**
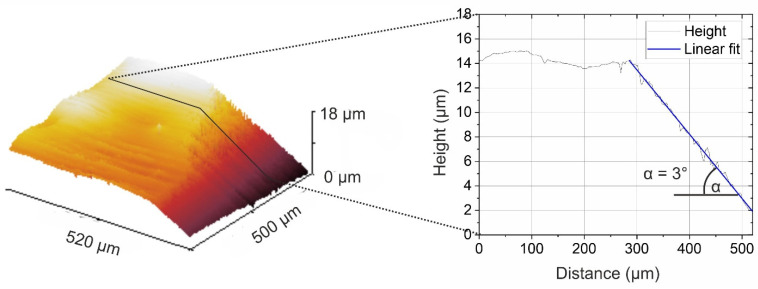
White light interferometric 3D image (**left**) and linescan (**right**) of the cut.

**Figure 7 polymers-14-02226-f007:**
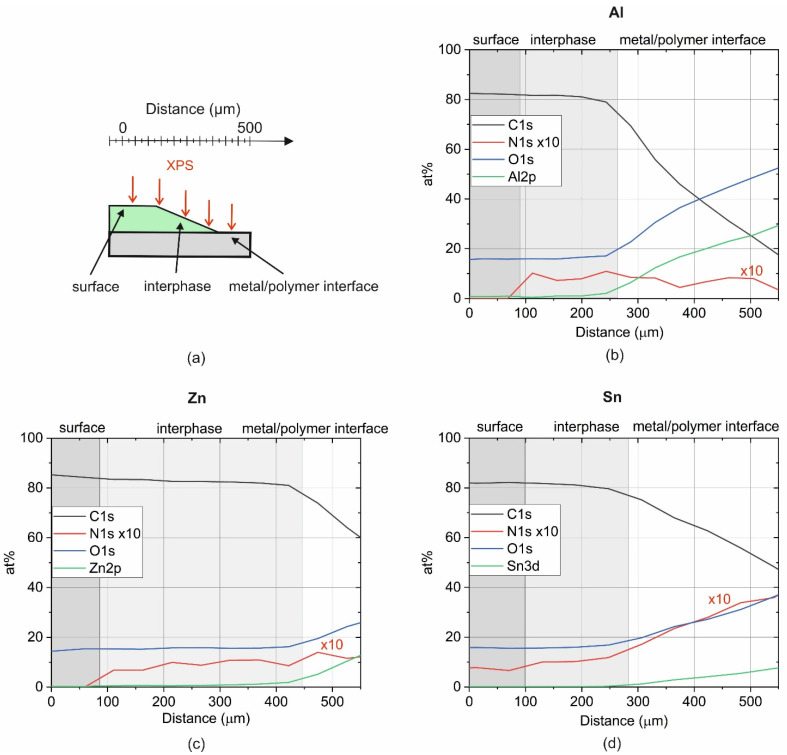
In (**a**), the schematic of an ULAM cut is depicted, along which XPS linescans have been performed, with (**b**) Al, (**c**) Zn, and (**d**) Sn as the substrate materials, respectively. The signal of N is multiplied by a factor of 10 in all graphs for better visibility. The light grey area labeled “interphase” represents the uncovered part of the coating with a thickness of 5 µm.

**Figure 8 polymers-14-02226-f008:**
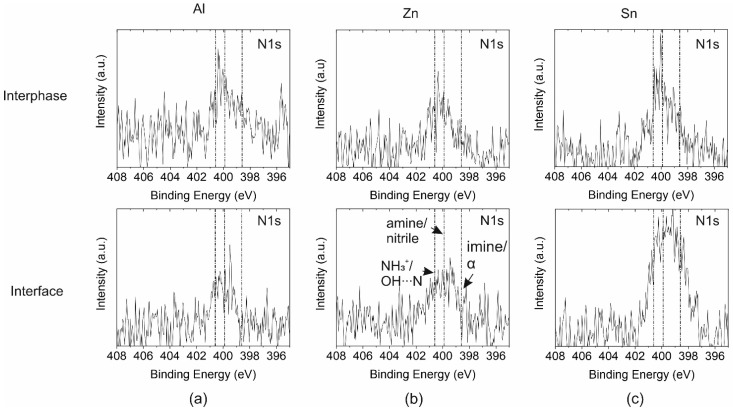
N1s peak in the interphase (i.e., the bulk of the epoxy coating) and at the epoxy–metal interface for (**a**) Al, (**b**) Zn, and (**c**) Sn.

**Table 1 polymers-14-02226-t001:** Binding energies and concentrations of evaluated functional groups of the components.

	C1s				N1s		O1s
	C-C	C-O	-C≡N (2)	C-N3 (1)	-C≡N/NH2 (3 + 5)	C=N-C (4)	C-O
Epoxy	285.0 (59.4 at%)	286.8 (24.4 at%)					533.2 (16.2 at%)
Emulsifier	285.0 (14.3 at%)	286.4 (57.4 at%)					532.6 (28.3 at%)
DICY	285.0 (4.7 at%)		287.0 (16.0 at%)	288.9 (16.4 at%)	399.9 (46.7 at%)	398.6 (16.2 at%)	

## Data Availability

Not applicable.

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
