# Peer review of "The Interaction of Waterborne Epoxy/Dicyandiamide Varnishes with Metal Oxides"

_polymers, 2022, doi:10.3390/polym14112226_

Round 1

Reviewer 1 Report

In this paper, authors elucidate potential interactions of waterborne epoxy varnishes based on the latent curing agent DICY and Al, Zn and Sn metal substrates. However, some issues should be clarified:

  1. In describing of“Materials and sample preparation”, schematics could be inserted.
  2. In Fig.5 and Fig.6, the images should be concise and no guide tieo is required
  3. It is recommended to add the references in the introduction parts (IEEE Transactions on Components, Packaging and Manufacturing Technology, 2016, 6(12): 1820-1826. IEEE Transactions on Components, Packaging and Manufacturing Technology, 2016, 6(9): 1317-1329.)

Reviewer 2 Report

The chemical reactions and interactions between epoxy/DICY and metal oxides (Al, Sn, Zn) were examined primarily based on XPS. The findings are important for some areas requiring interfaces between epoxy and metal substrates. 

  1. How did you determine the theoretical values of intensities?
  2. Please clarify the purpose of using hot water? Was it merely for dissolving DICY?
  3. Figure 2 is very hard to distinguish now. Please make clearly different colors for lines and use dashed and double-dashed lines in the figures.
  4. The authors investigated the interactions based on XPS. Thus, what were the suggested chemical structures of the complexes, based on the results? Please draw the chemical structures if applicable.
  5. In Figure 4, how to make sure if the phases were surfaces, interphases, and interfaces?
  6. Have the authors examined the curing behaviors and degrees? It may be different for metals.
  7. XPS is a powerful tool to study the chemical structures. However, have you considered additionally utilizing FTIR and DSC? Why didn't you use them?
  8. Please clarify the purposes of Figures 5 and 6. The audiences may not understand.

Round 2

Reviewer 2 Report

The revised manuscript substantially improved. It is worth publishing.